# A Fulminant Case of Adenovirus Genotype C108 Infection in a Pediatric Stem Cell Transplant Recipient with x-Linked Lymphoproliferative Syndrome Type 1

**DOI:** 10.3390/v16010137

**Published:** 2024-01-18

**Authors:** Bradford A. Becken, Daryl M. Lamson, Gabriel Gonzalez, Sachit Patel, Kirsten St. George, Adriana E. Kajon

**Affiliations:** 1Department of Pediatrics, University of Nebraska Medical Center, Omaha, NE 68198, USA; bbecken@childrensnebraska.org (B.A.B.); spatel@unmc.edu (S.P.); 2Wadsworth Center, New York State Department of Health, Albany, NY 12208, USA; daryl.lamson@health.ny.gov (D.M.L.); kirsten.st.george@health.ny.gov (K.S.G.); 3UCD National Virus Reference Laboratory, Dublin, Ireland; gabo.gonzalez@ivred.hokudai.ac.jp; 4Japan Initiative for World-Leading Vaccine Research and Development Centers, Institute for Vaccine Research and Development, Hokkaido University, Hokkaido, Japan; 5Lovelace Biomedical Research Institute, Albuquerque, NM 87108, USA

**Keywords:** adenovirus, transplant, HAdV-C, next generation sequencing

## Abstract

A 3-year-old male with X-linked lymphoproliferative syndrome type 1 underwent an unrelated umbilical cord blood transplant (UUCBT). The week prior to transplant the patient tested positive for adenovirus (HAdV) with a viral load of <190 copies/mL and was started on cidofovir. UUCBT proceeded as scheduled, and the patient engrafted on day +19. The patient’s HAdV load in serum continued to rise with resulting hepatic dysfunction, despite ongoing therapy with cidofovir and HAdV specific T-cell infusions. The patient died 6 months after transplantation having never cleared the virus. Next generation whole genome sequencing and sequence data analyses identified an intertypic recombinant HAdV-C P1H2F2 closely related (99.6% similarity) to genotype C108 in the isolates from three blood specimens obtained during the last week of life. Incidentally, the de novo assembly strategy enabled the detection of an adeno-associated virus type 2 (AAV2) genome in the DNA purified from the plasma isolates. Proteotyping analysis revealed minor differences in the predicted amino acid sequences for E1A, E1B 19K, E1B 55K, DNA polymerase, penton base, and fiber. None of the mutations previously described for HAdV-C5 variants resistant to cidofovir were identified. In silico restriction enzyme analysis revealed a distinct Sac I profile for the identified virus, supporting its designation as a C108 variant.

## 1. Introduction

Adenoviruses are non-enveloped, icosahedral, double stranded, linear DNA (dsDNA) viruses that affect a multitude of hosts and cause a wide range of illnesses in humans. The human adenoviruses (HAdVs) described to date have been classified into seven species, HAdV-A through HAdV-G, that collectively comprise 52 serotypes defined by their antigenic reactivities in neutralization assays and more than 114 unique genotypes defined by genomics criteria [1] (http://hadvwg.gmu.edu, accessed on 13 January 2024). In immunocompetent hosts, depending on the infecting species and type, infections can result in a variety of illnesses including pneumonia, conjunctivitis, gastroenteritis, hepatitis, myocarditis, pharyngitis, rash, and, rarely, encephalitis [1,2,3].

In the immunocompromised host, depending on age, HAdV infections can occur as a de novo/primary event or, most frequently, as a result of reactivation from latency. In this context, they can be associated with a wider spectrum of diseases including encephalitis, hepatitis, and hemorrhagic cystitis, as well as with higher mortality [2,4,5]. The severity and course of infection are likely dependent on a multitude of factors including virus type, but the availability of epidemiology data for this particular patient population is still very limited as typing is not carried out in most diagnosed cases of HAdV infection. A recent study by Grimley and colleagues showed HAdV-C to be the most frequently detected species among pediatric hematopoietic cell transplant recipients [6].

While HAdV infections are often self-limited in the immunocompetent host, treatment is frequently required in immunocompromised patients. Among the limited currently available options, the mainstay of antiviral therapy is Cidofovir, or the oral lipid ester form brincidofovir [6,7]. Reduced-intensity conditioning regimens [8], the prophylactic administration of intravenous immunoglobulin [9], and ex vivo-generated third-party or patient/specific donor-derived adenovirus-specific T cells (adenovirus VSTs) [10,11,12] are alternative approaches for viral control. A plethora of promising novel antiviral therapies, including DNA replication inhibitors and disruptors of viral gene expression, are currently under development [13].

Here we report the results of the virology investigation in a case of HAdV infection in a 3-year-old male who presented with a very low viremia immediately prior to an umbilical cord blood transplant for X-linked lymphoproliferative syndrome type 1 (XLP-1) and succumbed to the infection after enduring a high viral load for six months despite aggressive cidofovir and adenovirus specific T-cell infusion therapy. 

## 2. Materials and Methods

### 2.1. Virus Isolation and Initial Molecular Typing

The clinical specimens available for virology studies were limited to a serum sample and the corresponding clot from a week before the patient died, together with two heparin plasma samples and an EDTA-whole blood sample collected over the patient’s final two days. At Lovelace Biomedical Research Institute, all specimens were processed and inoculated on A549 cells in shell vials as previously described [14,15]. Isolates were further amplified for two additional passages for the extraction of highly pure intracellular viral DNA as previously described [16]. DNA preparations were examined by horizontal gel electrophoresis after digestion with *Bam* HI for a preliminary species identification. Initial molecular typing was carried out by PCR amplification and the Sanger sequencing of the hexon hypervariable regions (HVR) 1–7 and the fiber gene as previously described [17,18].

### 2.2. Next Generation Whole Genome Sequencing, Genome Assembly and Annotation

Purified viral DNA preps were processed for next generation sequencing in an Illumina Miseq instrument, as previously described [19]. For each sample the raw fastq reads were paired, merged, trimmed, normalized, and error corrected before de novo assembly in Geneious Prime version 2021.1.1 (Dotamatics, www.geneious.com). The resulting sequences were further annotated using VAPiD [20] and uploaded to GenBank using the Bankit tool. 

### 2.3. Sequence Data Analysis

The novel whole genome sequences were compared to the following genomic sequences available in GenBank for other *Human mastadenovirus* C genotypes: 1 (AC_000017), 2 (AC_000007), 5 (AC_000008), 6 (HQ413315), 57 (HQ003817), 89 (MH121097), 104 (MH558113), 108 (MF315029), 108* (OQ518339), 108** (OQ518330), and an intertypic recombinant genotype with a non-designated (ND) ID number (MF315028). 

Maximum-likelihood phylogenetic trees for the complete genome and open reading frames for penton base, hexon, and fiber were inferred using iQTree v2.0 [21,22,22]. The best fit model was determined in iQtree using the function -m TEST. The program chose the following parameters for the best tree: the nucleotide substitution model used general time reversible, GTR [23]; the frequency used empirical base frequency, F; the rate heterogeneity across sites allowed for a proportion of invariable sites, I, and the discrete gamma distribution with default four rate categories (G4) [24]; with standard non-parametric bootstrap at 100 repetitions; the command used was GTR + F + G4 − b 100.

For the evaluation of the similarity of the sequenced genomes to those of previously described genotypes, a multiple sequence alignment was built using MAFFT with the FFT-NS-I [25], and a sliding window analysis was performed with Simplot V3.5.1 [26] using a 500-nucleotide sliding window and 50-nucleotide step size, using GapStrip, on a Kimura distance model, and Ts/Tv = 2.0. The analysis and graphical representation of the results was performed with R version 4.0.5 [27]. The percentage of similarity of the novel sequences against the entire panel of genotypes in a pairwise fashion was carried out in MEGA7 v7.0.25 [28] with complete deletion for sites with gaps or missing data.

In silico restriction enzyme analysis was carried out in Geneious Prime 2021.1.1. The HAdV-C genomic sequences listed above were included in the comparative analysis. A large panel of endonucleases was selected for screening to identify those yielding profiles that could distinguish the most closely related genomes.

### 2.4. Proteotyping

The amino acid differences in the predicted sequences of E1A, E1B, DNA polymerase, penton base, hexon, and fiber proteins among all known HAdV-C genotypes were analyzed and visualized with the method of proteotyping adapted from Obenauer et al. [29]. The corresponding sequences were aligned then grouped according to their respective maximum likelihood phylogenetic tree, and the amino acid sites per type were colored according to the frequency of the different amino acids relative to the most frequently occurring residues for each site as previously described [30].

## 3. Results

### 3.1. Case Report

A 3-year-old male with no significant past medical history was diagnosed with X-linked lymphoproliferative syndrome type 1 (XLP-1). The patient had undergone genetic testing following a younger sibling’s diagnosis of Burkitt’s lymphoma. Despite the patient’s underlying diagnosis, prior to the planned unrelated umbilical cord blood transplant (UUCBT) he had done well and had not progressed to any of the feared complications of XLP-1, such as hemophagocytic lymphohistiocytosis (HLH), or developed Epstein–Barr virus (EBV) viremia. His only known prior infection had been respiratory syncytial virus (RSV), which had not resulted in hospitalization. While awaiting the transplant and subsequent engraftment, the patient was on a prophylactic regimen of acyclovir, pentamidine, and voriconazole to cover common infections for which the patient would be at risk given his immunocompromised state. One week prior to UUCBT, and after the initiation of his conditioning regimen, the patient was noted to have a positive HAdV serum PCR on routine monitoring, albeit below the quantifiable limit of detection of the test, <190 genome copies/mL. Conditioning was completed with a reduced-intensity regimen with alemtuzumab, fludarabine, melphalan, and thiotepa. Seven days after the initial positive HAdV PCR test, the patient underwent transplant and engrafted on day +17. 

Five days following the transplant, the HAdV load in the patient’s blood increased to log_10_ 5.31 (206,000) genome copies/mL, and he was started on weekly cidofovir, dosed at 5 mg/kg, as well as on probenecid. Despite therapy with cidofovir and engraftment on day +17, the adenovirus serum load further increased to log_10_ 7.88 copies/mL. Given the increasing viremia, the patient received his first infusion of allogenic HLA class II-matched adenovirus VSTs produced by stimulation with MACS^®^ GMP PepTivator AdV Select consisting of MHC class I and II oligopeptides derived from various proteins of HAdV-C2 and -C5 (https://www.miltenyibiotec.com/US-en/products/macs-gmp-peptivator-adv-select.html, accessed on 13 January 2024). The patient eventually received monthly adenovirus VST infusions, five doses in total, and remained on cidofovir, with dosing briefly changed to 1 mg/kg thrice weekly. HHV6 viremia was detected on day +39, and BK virus-associated cystitis was detected on days +60 and +67 accompanied by hematuria. Even with adenovirus VSTs and cidofovir therapy, the patient’s serum viral load never declined below log_10_ 6.7 genome copies/mL (Figure 1). 

Prior to the fourth infusion of adenovirus VSTs, the patient developed abdominal distension, transaminitis (AST/ALT 536/189), and jaundice (total bilirubin 14.1 mg/dL). The patient underwent an abdominal ultrasound showing complex ascites. Coupled with his clinical decompensation, a paracentesis was performed with cultures positive for an extended-spectrum beta-lactamases (ESBL)-producing *Klebsiella aerogenes,* prompting treatment with meropenem. A liver biopsy performed on day +91 and processed for histopathology examination revealed expanded portal triads with necrosis, cholestasis, and increased lymphocytic inflammation. Hepatocytes with nuclear viral cytopathic effects were present surrounding the areas of necrosis, highlighted by the immunochemical detection of HAdV (Figure 2). HAdV levels peaked at log_10_ 10.11 (12,882,495) copies/mL on day +154. Following the fifth T-cell infusion, the patient developed worsening fevers and hypotension. Blood cultures the day prior to the patient’s death grew meropenem-resistant *Pseudomonas aeruginosa.* Despite dual therapy, the patient died on day +170 without having cleared the HAdV infection (Figure 1). 

### 3.2. Virology Investigation and Findings

Adenovirus isolates were readily recovered from the only samples available for virology investigation: a blood clot specimen collected a week before the patient’s death, as well as two plasma specimens collected two days before the patient’s death. An initial molecular typing of the isolates by PCR amplification and Sanger sequencing of the hexon and fiber genes identified the detected virus as species HAdV-C with a type 2 hexon gene (H2) and a type 2 fiber gene (F2). 

To enable the comprehensive genomic characterization of the isolated viruses, whole genome sequences were obtained for the blood clot isolate and for one of the plasma isolates with an average depth of coverage of 29.1 and 61.8, respectively. An initial BLAST analysis of these two whole genome sequences confirmed the isolates to belong to species HAdV-C and identified them as identical intertypic recombinant genomes with a C1-like penton base gene (P), a C2-like hexon gene (H), and a C2-like fiber gene (F) (P1H2F2). The phylogenetic analysis of evolutionary relationships and sequence similarities to other HAdV-C genotypes (Figure 3 and Figure 4 and Table 1) identified the clinical isolates to be closely related to genotype C108 (with 99.6% seq similarity) as well as to a non-designated HAdV-C genotype (C ND, GenBank accession no. MF315028, with 99.1% sequence similarity) described by Mao and colleagues for viruses detected in cases of pediatric respiratory disease in Beijing, China in 2013 and 2012, respectively, and exhibiting the same molecular identities for the penton base, hexon, and fiber genes [31]. Importantly, the same high sequence similarity was found between the the genomes of the UNMC isolates and the genomes of two USA clinical isolates from 2009 and 2011 uploaded to GenBank in November 2023 (Accession no. OQ518339 and OQ518330, respectively). 

Simplot analysis of regions of similarity/divergence between the UNMC strain and the most closely related genotypes C108, C108*, C108**, C ND, and C2 revealed higher evolutionary diversity to these reference sequences in the E1A, E1B, E2B, and E4 coding regions. Interestingly, the evolutionary diversity was not consistent across the genomes of the three sequences for C108 isolates included in the analysis (C108, C108*, and C108**), providing evidence of considerable intratypic genetic variability.

As shown in Figure 5, in silico restriction enzyme analysis confirmed the similarity with the examined reference viruses and revealed a distinct profile for the UNMC isolates only with endonuclease *Sac* I. Based on these findings, and the current criteria for genotype designation (http://hadvwg.gmu.edu, accessed on 13 January 2024), the viruses isolated from the clot and plasma specimens were identified as a variant of genotype C108 and designated HAdVC/USA/UNMC-c/2021/108v[P1H2F2] and HAdVC/USA/UNMC-p/2021/108v [P1H2F2], respectively. 

Proteotyping analysis for predicted polypeptides encoded in E1A, E1B, DNA polymerase, L2 Penton base, L3 Hexon, and L5 Fiber revealed some unique features for the UNMC strain (Figure 6A,B). A closer examination of the predicted amino acid sequence for the DNA Polymerase revealed none of the amino acid changes previously described in positions 87 and 303 for brincidofovir-resistant HAdV-C5 variants selected in vitro by an extended passage of wild-type HAdV-C5 in the presence of brincidofovir [32]. Importantly, as shown in Figure 6A, all of the HAdV-C genotypes included in our analysis displayed identical sequences in these two positions.

Interestingly, our sequencing protocol and de novo contig assembly strategy identified the presence of an adeno-associated virus type 2 (AAV2) genome in the purified DNA preps from the two plasma isolates. 

Whole genome sequences were deposited in GenBank under accession numbers OQ108498 and OQ108499 for the plasma and clot HAdV-C108v isolates, and OQ130187 for the complete genomic sequence of the detected co-infecting AAV2 designated UNMC-P-0424_AAV2.

## 4. Discussion

XLP-1 syndrome is an extremely rare inherited (primary) immunodeficiency disorder that affects one to three men per million worldwide and is caused by a variety of mutations in the *SH2D1A* gene, encoding signaling lymphocytic activation molecule (SLAM)-associated protein (SAP) [33]. The syndrome is characterized by a defective immune system that is hyperresponsive to infection, but no clear correlations between individual mutations and the severity of the associated phenotypes have been established. Epstein–Barr Virus (EBV) infection is a well-identified trigger of immune dysregulation and HLH in XLP-1 patients [34,35], but overall, little is known about the pathological consequences of other viral infections common in childhood in individuals with this disorder. Interestingly, a case of fatal disease associated with SARS-CoV-2 infection was recently reported in an XLP-1 patient by Chung et al. [36]. Like with EBV, HAdV infection may have a significant impact on children with primary immunodeficiencies and may be an underestimated cause of morbidity in these patients.

Based on the virology data available for our case from the weekly monitoring of viral load in blood by qPCR, it is clear that shortly after the initiation of his conditioning regimen the patient had low-level but detectable HAdV viremia that escalated during the course of the subsequent 30 weeks and could not be controlled by the combined cidofovir therapy and adenovirus VST infusions. Whether this was a consequence of a primary infection or of a reactivation of a latent virus, and the reasons underlying the failure of treatment, will be impossible to establish with the limited information available.

The novel intertypic recombinant HAdV-C strain isolated from three independent blood specimens obtained from the patient within his last week of life is most closely related to genotype C108, with which it shares the highest percentage of whole genome sequence similarity (99.6%) and the molecular identity of the three loci used in molecular typing, P1 H2 F2. Genotype C108 was originally described for strain BJ09 isolated from a respiratory specimen obtained in Beijing in 2013 from an infant with mild acute respiratory illness [31,37] and interestingly also reported to GenBank in November 2023 by Sereewit and colleagues for various USA isolates obtained between 2009 and 2016, including the ones we selected for the presented analysis (accession no. OQ518339 and OQ518330).

Without examining the virus’s in vitro and in vivo growth and pathogenic phenotypes, it is hard to speculate about a possible higher virulence of the C108 variant isolated from our case. At the genomic level, this variant and the closely related C108*, C108**, and C ND differ from C108 most notably in their genetic make-up in the E1 and E4 regions, supporting the observations of Dhingra and colleagues [38] regarding the patterns of the molecular evolution of species HAdV-C. 

A significant limitation of the virology studies conducted to investigate this case is, without a doubt, the lack of a formal in vitro evaluation of the antiviral activity of cidofovir against the isolated viruses to help explain the apparent refractory nature of the detected HAdV infection. However, our comprehensive genonome sequence data analysis did allow us to make the important observation that the genomes of the isolated C108 variant viruses did not feature any of the mutations in the DNA Polymerase coding sequence positions previously reported in association with brincidofovir resistance in vitro [30].

Although possible, based on the recent findings in the context of acute severe hepatitis cases in children in Scotland and the US [39,40], the contribution of AAV2 co-infection to the severity of the case described in this manuscript is also hard to elucidate because the presence of AAV2 was only detected in blood specimens collected within the last two days of the patient’s life. The detected reactivation of HHV6 and BK virus and the two opportunistic bacterial infections likely also contributed to the poor outcome.

## 5. Conclusions

Whole genome sequencing and analyses are currently the most powerful tools for the characterization of HAdVs of medical importance and for the investigation of cases of special interest. The initial molecular typing was restricted to amplifying and sequencing the hexon HVR 1–7, and the fiber gene would have identified the isolated virus as HAdV-C2. The description by Mao et al. in 2017 of two closely related but distinct novel intertypic recombinants strains of HAdV-C with a type 1-like penton base gene (P1), a type 2-like hexon gene (H2), and a type 2-like fiber gene (F2) isolated from cases of pediatric respiratory illness in Beijing, China [31] supported the recognition of genotype C108 by the Human Adenovirus Working Grup. Taken together, the identification of a C108 variant in our 2021 XLP-1 case and the recent uploading to GenBank of various highly similar whole genome sequences corresponding to clinical isolates of unreported source obtained in the State of Washington, USA between 2009 and 2016 suggest that HAdV-C viruses with this particular molecular identity (P1H2F2) may be widely spread.

## Figures and Tables

**Figure 1 viruses-16-00137-f001:**
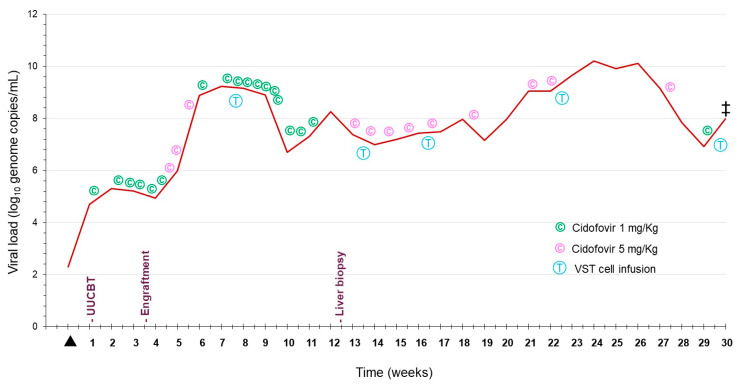
Serum adenoviral load, treatments, and procedures over the thirty weeks following the first HAdV positive test until the patient’s death. ▲ First HAdV-positive qPCR test; UUCBT: unrelated umbilical cord blood transplant; ^‡^ Patient death.

**Figure 2 viruses-16-00137-f002:**
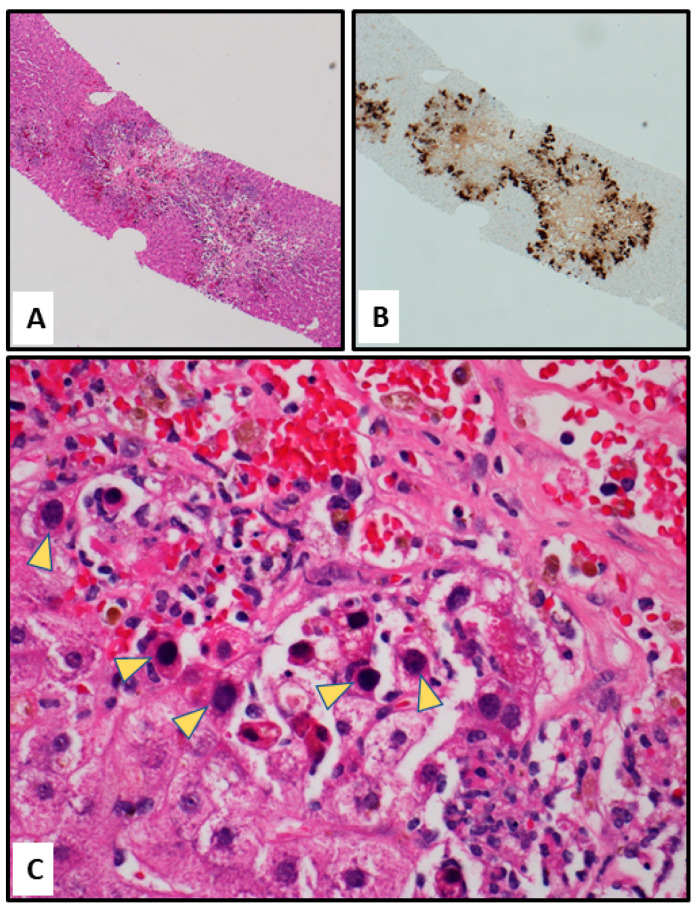
Histopathology of liver biopsy (day 91 post-transplant). (**A**) Hematoxylin and Eosin stain (H&E) at 4× magnification. (**B**) Immunohistochemistry for adenovirus at 4× magnification. (**C**) H&E stain at 40× magnification. Yellow arrows point at nuclei of infected cells.

**Figure 3 viruses-16-00137-f003:**
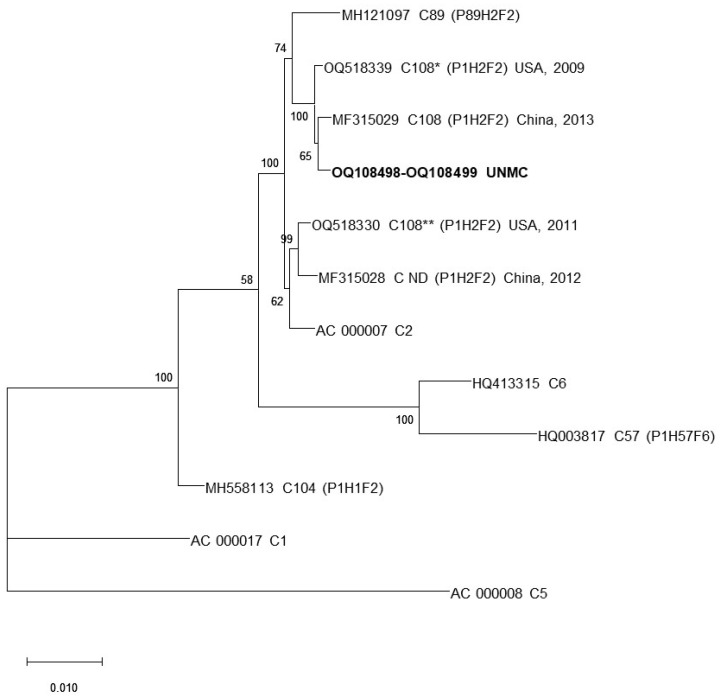
Phylogenetic analysis of whole genome sequences of UNMC isolates and closely related genotypes, C1, C2, C5, C6, C57, C89, C104, C108, and C-ND (genotype not formally designated). C108* and C108** designate the genome sequences of two clinical isolates reported to GenBank in November 2023. The maximum likelihood phylogenetic tree was inferred using iQtree v2.0. Values next to the branching nodes represent the bootstrap support, and the length of the branches represents the number of mutations per nucleotide site according to the scale. Bootstrap values < 70 are not shown.

**Figure 4 viruses-16-00137-f004:**
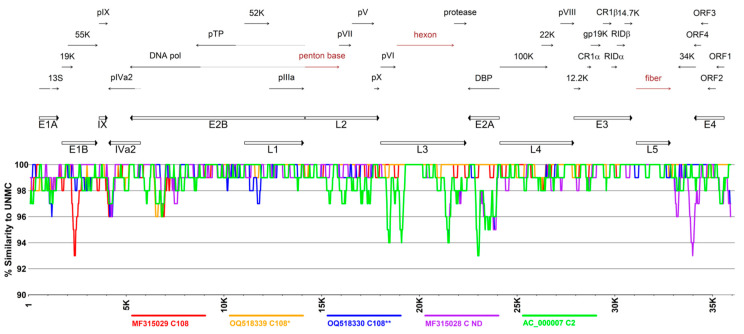
Analysis of whole genome sequences for regions of similarity and divergence between the UNMC isolate and the most closely related genotypes C108, C108*, C108**, C ND, and C2. The horizontal and vertical axes represent the genome positions and the percentage of similarity, respectively. The top panel presents the common annotation of *Human mastadenovirus* C genomes with coding regions for the major capsid proteins, penton base, hexon, and fiber indicated in red font. The bottom panel shows the similarity of the selected genomes to the genome of the UNMC clinical isolate. Series are colored according to the genotypes shown at the bottom of the panel.

**Figure 5 viruses-16-00137-f005:**
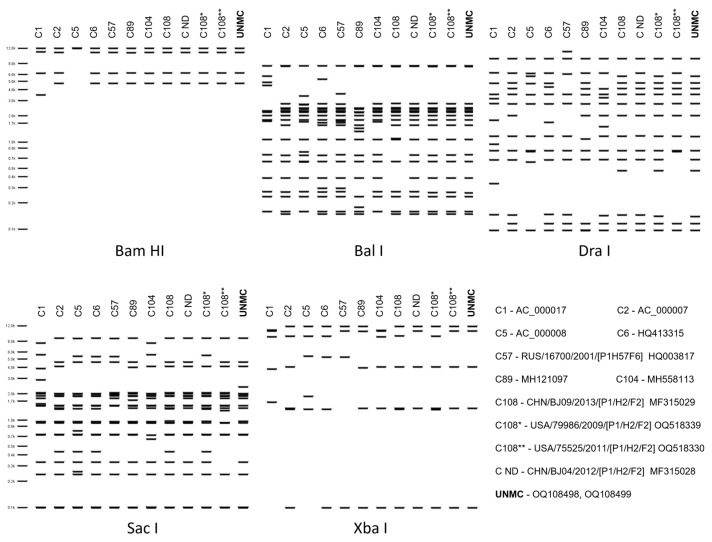
In silico restriction enzyme analysis of the genomes of UNMC isolates and closely related genotypes, C1, C2, C5, C6, C57, C89, C104, C108, C108*, C108**, and C ND (not designated) with endonucleases Bam HI, Bal I, Dra I, Sac I, and Xba I. The analysis was conducted in Geneious Prime.

**Figure 6 viruses-16-00137-f006:**
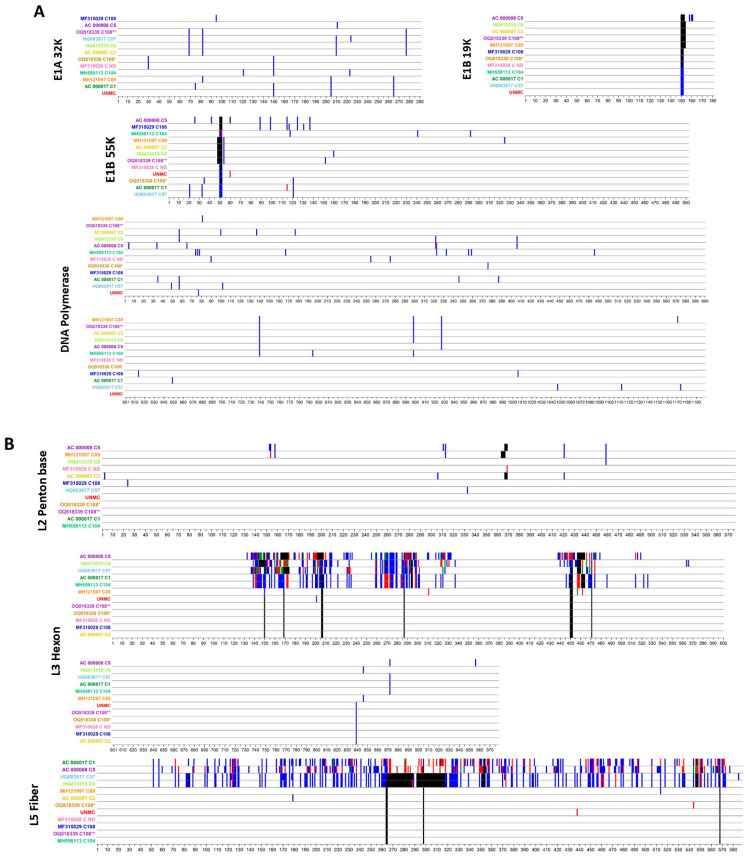
Proteotyping analysis. Vertical axes correspond to the compared genotype sequences (names on the left of the panel), and the horizontal axes correspond to amino acid positions in the predicted sequences for (**A**) the non-structural proteins E1A 32K, E1B 19K, E1B 55K, and DNA Polymerase, and (**B**) for the major capsid proteins Penton base, Hexon, and Fiber. Amino acid positions with polymorphisms are colored according to the frequency across sequences, with blank for the consensus, blue and red for the first and second most frequent polymorphisms for a given position, and green for the third most frequent amino acid for that position; gap sites are colored in black.

**Table 1 viruses-16-00137-t001:** Analysis of sequence similarities (%) of UNMC strain with all other known HAdV-C genotypes.

Genotype Accession No.	Seq	C1 AC000017	C2 AC000007	C5 AC000008	C6 HQ413315	C57 HQ003817	C89 MH121097	C104 MH558113	C108 MF315029	C108* OQ518339	C ND ^‡^ MF315028	C108** OQ518330
Molecular ID		P1H1F1	P2H2F2	P5H5F5	P6H6F6	P1H5F6	P89H2F2	P1H1F2	P1H2F2	P1H2F2	P1H2F2	P1H2F2
UNMC+ OQ108498/OQ108499	WGS	94.8	98.9	93.7	96.5	96	98.7	98.0	**99.6**	99.5	99.1	99.2
Penton base	Nt	**99.9**	98.7	98.0	99.7	**99.9**	97.7	**99.9**	99.8	99.9	99.8	99.9
	AA	100	99.0	98.3	99.8	99.8	98.3	100	99.8	100	99.8	100
Hexon	Nt	85.5	98.8	81.7	88.2	87.2	97.5	85.4	**99.9**	99.9	98.8	98.8
	AA	90.6	99.9	86.3	89.4	90.2	99.3	90.6	99.9	99.9	99.9	99.9
Fiber	Nt	72.1	99.7	72.6	67.7	67.3	99.6	99.8	**99.9**	99.8	99.8	99.7
	AA	72.5	99.7	69.2	67.2	66.4	99.7	99.8	99.8	99.8	99.8	99.7

* and **: genomic sequences for C108 strains uploaded in GenBank in November 2023. ^‡^: Genotype not formally designated. WGS: Whole genome sequence. Nt: Nucleotide. AA: Amino acid.

## Data Availability

All raw data are available upon request. Our genomic sequence data are available from NCBI/GenBank under accession numbers OQ108498, OQ108499 and OQ130187.

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
