# Peer review of "A Fulminant Case of Adenovirus Genotype C108 Infection in a Pediatric Stem Cell Transplant Recipient with x-Linked Lymphoproliferative Syndrome Type 1"

_viruses, 2024, doi:10.3390/v16010137_

Round 1

Reviewer 1 Report

Comments and Suggestions for Authors

This is a well written manuscript that covers an increasingly important topic on molecular diagnostics of adenoviral infections. The finding that restricted molecular analysis of only restricted viral genomic regions can lead to mistyping of the viral strain is of great importance to be able to drive diagnostic testing to full viral genome sequencing. Your identification of mutations that may alter the polypeptides encoded in the E1A, E1B, DNA pol, penton, hexon and fiber are interesting as other virulent isolates have mutations within some of the same regions. 

Author Response

We want to thank Reviewer 1 for the positive and encouraging comments on our manuscript

Reviewer 2 Report

Comments and Suggestions for Authors

Becken et al report the isolation and genomic characteristics of a HAdV isolate from a pediatric patient undergoing an umbilical cord blood transplant.  This is sort of a hybrid manuscript with features of a case report and genomic characteristics of a HAdV isolate that caused fatal disease.

The manuscript is clearly written and the figures appear well constructed and appropriate. The case described is quite complex, with several other infectious agents involved including two bacteria and AAV-2. 

As noted by the author's, an actual determination of whether the HAdV isolate is resistant to cidofovir would be a very welcome addition to the study, despite the evidence that several mutations associated with resistance are not present from the genome sequencing.  Given that this isolate is very similar to other reports from various distant locations, it could be very important to know if this virus is indeed resistant to cidofovir and what mutations contribute to resistance. Nevertheless, I am willing to consider this manuscript without that analysis.

Major point:

1) The viral specific T cell (VST) therapy is not described in any detail.  Given that this therapy failed, some details must be provided.  For example, was this autologous or derived from a third party? Was there an effort to match MHC-II?  What peptides were used for expansion?  Where they from HAdV-C, types 1 or 2, etc?

Minor point:

Although well referenced, there are a number of newer reviews/papers in the field that might also be worth referencing, including the pandemic/disease potential of HAdV (PMIDs: 36851544; 34473804; 36336610), anti-adenoviral therapies (PMID: 33577808) and adenoviral VST therapy specifically (PMID: 344732370).

Author Response

We want to thank the reviewer for positive comments on our manuscripts and for the suggestions for improvement. We have edited the manuscript to address the comments and requests for additional information as follows:

Major point: The viral specific T cell (VST) therapy is not described in any detail.  Given that this therapy failed, some details must be provided.  For example, was this autologous or derived from a third party? Was there an effort to match MHC-II?  What peptides were used for expansion?  Where they from HAdV-C, types 1 or 2, etc?

We have edited the text on lines 142-145 to provide as much information as available

Minor point: Although well referenced, there are a number of newer reviews/papers in the field that might also be worth referencing, including the pandemic/disease potential of HAdV (PMIDs: 36851544; 34473804; 36336610), anti-adenoviral therapies (PMID: 33577808) and adenoviral VST therapy specifically (PMID: 344732370).

We have edited the text on line 59 to incorporate reference PMID: 36736781 [12], and on line 62 to incorporate reference PMID: 33577808.     

Because we could not find any reference matching the PMID number suggested/provided by the reviewer we have incorporated a very relevant and recent reference as the new reference 12 (PMID: 36736781).

We hope these edits meet the expectations to find our paper acceptable for publication.

Sincerely,

Adriana E. Kajon and co-authors